# A Case Report of Left Atrial Isomerism in a Syndromic Context

**DOI:** 10.3390/genes11101211

**Published:** 2020-10-16

**Authors:** Aurora Ilian, Andrei Motoc, Ligia Balulescu, Cristina Secosan, Dorin Grigoras, Laurentiu Pirtea

**Affiliations:** 1Department of Obstetrics and Gynecology, University of Medicine and Pharmacy “Victor Babes”, 300041 Timisoara, Romania; aurora1985@yahoo.com (A.I.); cristina.secosan@gmail.com (C.S.); grigorasdorin@ymail.com (D.G.); laurentiupirtea@gmail.com (L.P.); 2Department of Anatomy and Embryology, University of Medicine and Pharmacy “Victor Babes”, 300041 Timisoara, Romania; amotoc@umft.ro

**Keywords:** microarray, isomerism, syndrome

## Abstract

The objective of our paper is to underline the importance of assessing microarray genetic analysis for the detection of chromosomal abnormalities in rare cases such as left atrial isomerism, mostly in the context of antenatally detected syndromes. We present the case of a 26-year-old primipara, at 26 weeks of gestation, with prior first trimester normal anomaly scan, who presented in our department accusing lower abdominal pain. An anomaly ultrasound examination of the fetus revealed cardiomegaly with increased size of the right atrium, non-visualization of the atrial septum or the foramen ovale, malalignment of the three-vessel view, location of the superior vena cava above the two-vessel view, slight pericardial effusion, and no interruption of the inferior vena cava nor presence of azygos vein being noted. Associated extracardiac abnormalities, such as small kidneys at the level of the iliac fossa, micrognathia, dolichocephaly with hypoplasia of the cerebellum, increased nuchal fold, and reduced fetal movements were also reported. A diagnostic amniocentesis was performed, and, while the conventional rapid prenatal diagnostic test of the multiplex quantitative fluorescent polymerase chain reaction (PCR) came as normal, the microarray analysis (ChAS, NCBI Built 37 hg 19, detection of microdeletions or microduplications larger than 100 kb) revealed two chromosomal abnormalities: a 22.84 Mb loss of genetic material in the 18q21.31–18q23 chromosomal region and a gain of 22.31 Mb of genetic material in the 20p13–20p11.21 chromosomal region. After the termination of pregnancy, a necropsy of the fetus was performed, confirming heterotaxy syndrome with a common atrium, no atrial septum, superior vena cava draining medianly, and pulmonary veins that drained into the lower segment of the left atrium due to an anatomically enlarged single common atrium. The extracardiac findings consisted of two bilobar lungs, dysmorphic facies, low-set ears, nuchal fold edema, and small kidneys located in the iliac fossa. These findings are conclusive evidence that left atrial isomerism is a more complex syndrome. The genetic tests of the parents did not reveal any translocations of chromosomes 18 and 20 when the Fluorescent in situ Hybridization (FISH) analysis was assessed. The antenatal detection of corroboration between different structural abnormalities using serial ultrasound examinations and cardiac abnormalities, together with the detection of the affected chromosomes, improves the genetic counseling regarding the prognosis of the fetus and the recurrence rate of the condition for siblings.

## 1. Introduction

Fetal heterotaxy syndrome, also known as situs ambiguous, is a general term used to describe a wide correlation of conditions associated with abnormal organ arrangement. Previously defined as asplenia and polysplenia, pathologists observed that these conditions are better classified by describing the atrial morphology rather than describing the abdominal organs. Thus, the terms right and left isomerism have started to be introduced and used [1]. Left atrial isomerism is associated with the “double” left-sided structures and the underdevelopment or absence of the right-sided structures [1,2]. The extracardiac malformations primarily involve the abdominal cavity, and anomalies of the face, limbs, and brain can occur but are not typical. The most severe extracardiac malformation is extrahepatic biliary atresia with the absence of the gallbladder [3,4]. With our case report presentation, we want to underline the importance of ultrasound detection of the various extracardiac malformations associated with left atrial isomerism in combination with the microarray diagnostic test analysis, a technology that examines the genome with increased resolution [5]. Even though it is known that chromosomal aberrations are rare in left atrial isomerism, the inclusion of a wider spectrum of anomalies can detect chromosomal abnormalities and improve the couple’s counseling and decision regarding the prognosis and outcome of the pregnancy [5]. We present the case of a 26-year-old primiparous woman with left atrial isomerism that is part of a wider syndrome that implies two chromosomal abnormalities that were detected with the microarray analysis (ChAS, NCBI Built 37 hg 19). They were described as a 22.84 Mb loss of genetic material in the 18q21.31–18qq23 chromosomal region and a gain of 22.31 Mb of genetic material in the 20p13–20p11.21 chromosomal region. Both genomic sites involved were described by the geneticist to have a poor prognosis regarding quality of life due to the short stature of affected siblings, mental retardation, learning difficulties, and facial structural defects.

Written informed consent was obtained from the patient before her inclusion in the study. All procedures were performed in accordance with the ethical standard laid down in the 1964 Declaration of Helsinki and its later amendments and were approved by the Institutional Review Board and Ethical Committee of the “Victor Babeș” University of Medicine and Pharmacy, Timișoara. 

The study was registered by the Institutional Review Board and Ethical Committee of the “Victor Babeș” University of Medicine and Pharmacy, Timișoara, under the registration number 121/2020.

## 2. Case Presentation

We present the case of a 26-year-old primipara, at 26 weeks of gestation, with prior first trimester normal anomaly scan, who presented in our department with a complaint of lower abdominal pain. From her obstetrical history, the pregnancy was classified as low obstetrical risk, with a low risk for chromosomal anomalies at the first trimester scan, a normal anomaly scan, and normal family and personal history. An anomaly ultrasound examination of the fetus revealed cardiomegaly with an increased size of the right atrium, non-visualization of the atrial septum or the foramen ovale (Figure 1A–C), malalignment of the three-vessel views, location of superior vena cava above the two-vessel view, and slight pericardial effusion (Figure 2A,B). The associated extracardiac abnormalities included small kidneys located at the level of the iliac fossa, micrognathia (Figure 3A–C), dolichocephaly with hypoplasia of the cerebellum, increased nuchal fold (Figure 4A–C), and reduced fetal movements. The ultrasound cardiac findings were confirmed by a pediatric cardiologist and a diagnostic amniocentesis was performed.

Regarding the cardiac relations and the segmental approach, we noted an “A” situs ambiguous of a left atrial isomerism with a left atrial appendage and a rightward rotation with a D-loop of the ventricular mass (right hand pattern ventricular topology “D”) together with “S” normally related arteries (“ADS”) [6,7]. Mixed biventricular atrioventricular connections guarded by two normal atrioventricular (AV) valves, the mitral valve at the level of the left side of the heart and the tricuspid valve located at the level of the right side of the heart. Concordant ventriculoarterial connections (in biventricular heart) and normal position of the heart within the chest, with a left-sided ventricular mass and a left-sided cardiac apex were detected. A normal relationship was noted between the great vessels and descending aorta, with a left-sided descending aorta and a right-sided inferior vena cava in relation to the spine, as well as a normal relation regarding the aortic orifice, situated right and posterior with respect to the pulmonary valve. A particularity of the superior vena cava that was located above the plane of the three-vessel view was noted (Figure 2A), a deviation determined by the enlarged right-sided atrium. All four systemic veins drained into the left atrium with the anatomically normal systemic veins draining into the right-sided atrial chamber that had a morphologically left appendage. Within the heart, no atrial septum or foramen ovale was noted. We noted the presence of the gallbladder, one anatomically normal spleen, normal splenic artery, and a normally structured and positioned fetal liver in the right side of the abdomen.

The fetal karyotype was obtained in utero and the polymerase chain reaction (PCR) test result came as normal, with a microarray analysis revealing a male profile with two chromosomal abnormalities. Regarding results, the rapid prenatal diagnosis of the multiplex quantitative fluorescent polymerase chain reaction (PCR) test came as normal, while the microarray analysis (ChAS, NCBI Built 37 hg 19) revealed two chromosomal abnormalities: a 22.84 Mb loss of genetic material in the 18q21.31–18qq23 chromosomal region and a gain of 22.31 Mb of genetic material in the 20p13–20p11.2 chromosomal region. The genetic report described the 18q deletion syndrome as associated with short stature, mental disability, learning difficulties, hypotony, hearing loss, microcephaly, cleft palate, feet abnormalities, and congenital cardiac abnormalities, and the detected 20p duplication chromosomal abnormality is also associated with short stature, mental disability, learning difficulties, dysmorphic features, multiple congenital abnormalities, and autism. The couple decided to terminate the pregnancy due to the poor prognosis given by the association of multiple anomalies and the chromosomal abnormalities. After the termination of pregnancy, a necropsy of the fetus was performed, confirming heterotaxy syndrome with a common enlarged atrium (Figure 5B), no atrial septum, superior vena cava draining medianly, and pulmonary veins that drain into the lower segment of the atrium due to an anatomically enlarged single common atrium. The extracardiac findings consisted of two bilobed (left) lungs (left isomerism) (Figure 5A), dysmorphic facies, low-set ears, nuchal fold edema and small kidneys located in the iliac fossa. These findings are conclusive evidence that left atrial isomerism is a more complex syndrome. The genetic tests of the parents did not reveal any translocations of the chromosomes 18 and 20 when the Fluorescent in situ Hybridization (FISH) analysis was assessed. The antenatal detection of corroboration between different structural abnormalities using serial ultrasound examinations and cardiac abnormalities, together with the detection of the affected chromosomes, improves the genetic counseling regarding the prognosis of the fetus and the recurrence rate of the condition for siblings [5].

## 3. Discussion and Conclusions

This current report presents a case of fetal left atrial isomerism that is part of a syndromic context. The two chromosomal abnormalities were detected with the microarray analysis (ChAS, NCBI Built 37 hg 19), being described as a 22.84 Mb loss of genetic material in the 18q21.31–18qq23 chromosomal region and a gain of 22.31 Mb of genetic material in the 20p13–20p11.21 chromosomal region. The condition was identified later in pregnancy due to follow-up scans that revealed severe structural abnormalities, such as hypoplastic kidneys and cerebellar hypoplasia. Regarding the severity of the condition’s spectrum, it has been reported that heterotaxy syndrome is associated with various mutations in the primary ciliary dyskinesia gene site, and these features give an increased risk of postnatal respiratory complications due to the ciliary dysfunction [1,4,8,9].

Recent studies found that approximately 6% of patients with primary ciliary dyskinesia have heterotaxy. Kennedy et al. explained that motile cilia are required for the normal development of left–right asymmetry [10].

Laterality defects (including situs inversus totalis and, less commonly, heterotaxy, and congenital heart disease), are diagnosed in almost half of the patients with primary ciliary dyskinesia, the high incidence reflecting the dysfunction of embryological nodal cilia [10,11].

The incidence of trisomies in left atrial isomerism is rare, with occasionally reported 22q11 microdeletion [1,12]. Good prognosis is reported for newborns with antenatally mild, isolated cardiac abnormalities with present gallbladder, no heart block, and no other associated structural abnormality. 

A case of heterotaxy syndrome where microarray comparative genomic hybridization (CGH) detected an 18q.22.1q23 Hg 19 copy number variation (CNV) deletion syndrome was reported in a study design concerning intestinal malrotation. The reported associated clinical features included an accessory spleen on the descending colon, left-sided clubfoot, umbilical hernia, and congenital nystagmus with delayed myelination as neurological deficits [13].

There are no cases mentioned in the literature associated with fetal heterotaxy syndrome and duplications in the region of chromosome 20. 

Very few cases of 20p duplication have been reported. The majority of those were partial duplication (involving one part of the p arm) and occurred as part of a translocation (along with a deletion on another chromosome) [14]. Therefore, specifically due to the duplication, it is hard to know which symptoms were encountered in people with a 20p. Signs and symptoms reported in people with 20p duplication include intellectual disability, developmental delay, speech delay, poor coordination, dental problems, spine bone abnormalities, distinctive facial features, and heart problems.

## 4. Conclusions

We consider that microarray analysis should be offered when a suspicion of fetal heterotaxy is raised, especially when other structural abnormalities are noted antenatally on the ultrasound examination. Given the low number of cases of 20p duplication reported, it is difficult to evaluate the possible link between left atrial isomerism and the 18q deletion and 20p duplication. Our case presented these modifications and could be the starting point for further studies in this direction. Furthermore, the necropsy examination can confirm the various aspects of fetal heterotaxy syndrome and guide the genetic counseling, the future plan of conceiving, and genetic investigations.

## Figures and Tables

**Figure 1 genes-11-01211-f001:**
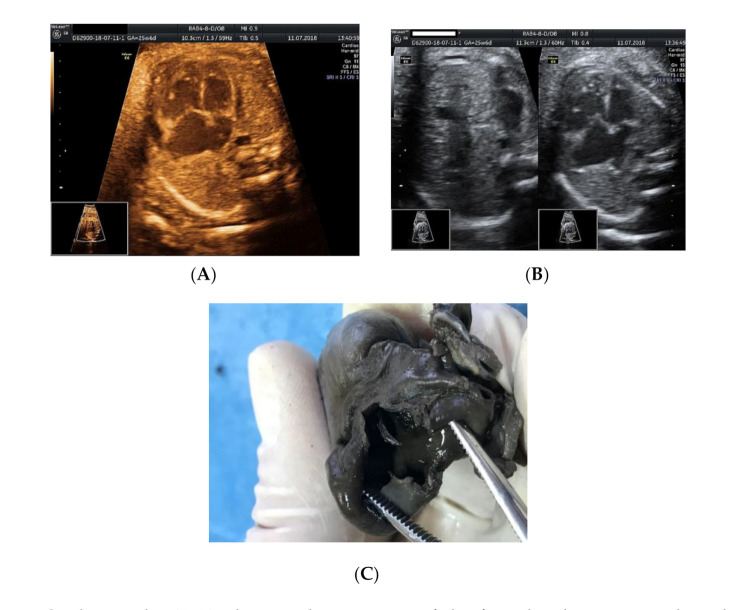
Cardiomegaly: (**A**,**B**) ultrasound assessment of the four-chamber view—enlarged right atrium, no foramen ovale noted, no atrial septum; (**C**) necropsy examination revealed one common atrium and no interatrial septum.

**Figure 2 genes-11-01211-f002:**
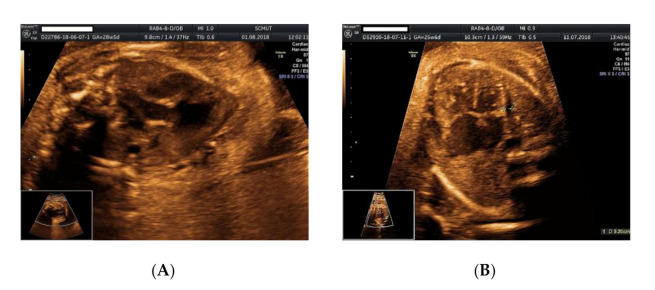
Ultrasound examination of the fetus: (**A**) malalignment of the three-vessel view with vena cava located above the aforementioned plane, enlarged right atrium, no foramen ovale, no interatrial septum; (**B**) pericardial effusion, no atrial septum, no patent foramen ovale noted.

**Figure 3 genes-11-01211-f003:**
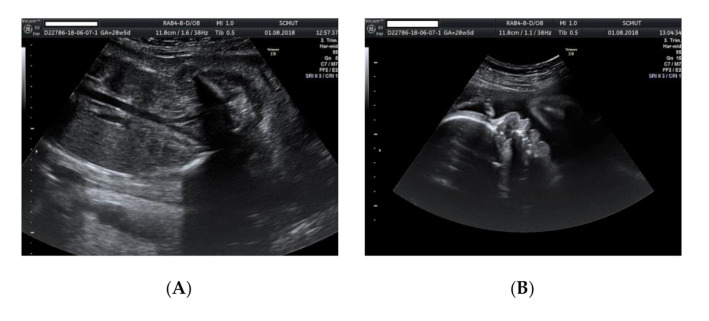
Ultrasound examination of the fetus: (**A**) small kidneys located in the iliac fossa; (**B**) micrognathia (2D image); (**C**) micrognathia (3D image).

**Figure 4 genes-11-01211-f004:**
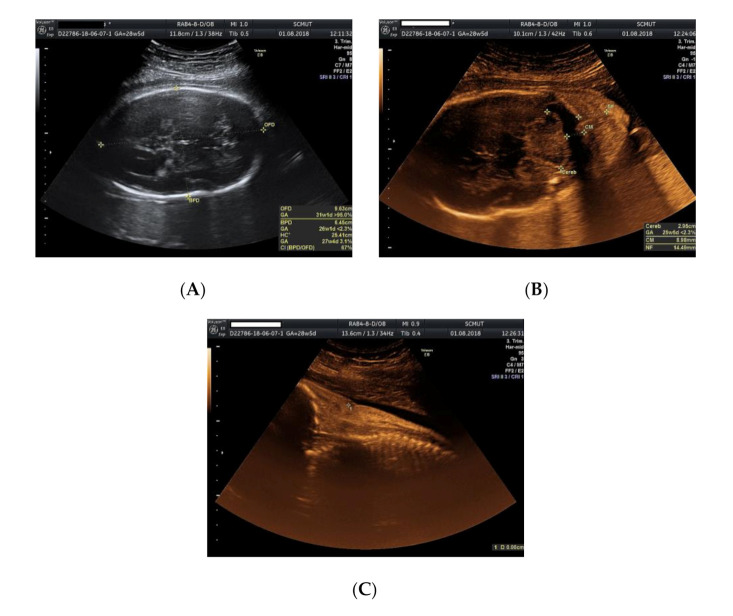
Ultrasound examination of the fetus: (**A**) dolichocephaly; (**B**) cerebellar hypoplasia; (**C**) increased nuchal fold.

**Figure 5 genes-11-01211-f005:**
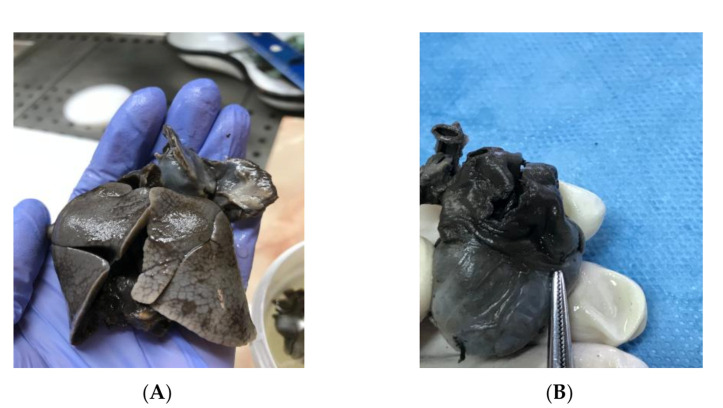
Macroscopical examination of the fetal heart and lungs during necropsy: (**A**) left atrial isomerism and bilobar lungs; (**B**) enlarged common atrium.

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
