# Peer review of "A Case Report of Left Atrial Isomerism in a Syndromic Context"

_genes, 2020, doi:10.3390/genes11101211_

Round 1

Reviewer 1 Report

The paper is well written and interesting. The topic is original and quality of presentation is good. Scientific approach is clear, figures are easy to understand and Bibliography is up to date. Conclusions are consistent with
evidence and arguments well presented.

Author Response

Dear reviewer,

Thank you for taking the time to evaluate our manuscript.

We very much appreciate your king remarks and the interest in our case report.

Best regards,

Ligia Balulescu

Reviewer 2 Report

Ilian and colleagues describe a case of left atrial isomerism in a fetus with polymaformations and 2 chromosomal aberrations (18q deletion and 20p duplication). The case is quite interesting. However, there are many aspects of the paper that should be addressed.

In particular:

  • Due to the fact that heterotaxy syndrome is a very complex condition, characterized by severe cardiovascular malformations, the authors describe only the morphology of the atrial appendages and the presence of a common atrium. One of the landmarks of the left isomerism is the interruption of hepatic portion of the inferior vena cava with azygos continuation: the authors do not describe the inferior vena cava at all. Moreover, the authors do not describe the ventricular loop (L- o D-loop?), the atrioventricular connection, the ventriculoarterial connection or the presence of other structural malformations of the heart.
  • Line 74. The sentences: “Superior vena cava draining medianly, the pulmonary veins that drain in the lower segment of the atrium” are confusing. The anatomic description is very imprecise.
  • Line 109. The link between mutations in the primary ciliary dyskinesia genes and heterotaxy syndrome should be expanded.
  • Lines 112-118. The sentences reported in these lines are very confusing and it is very difficult to understand the meaning.
  • The 18q deletion and 20p duplication of the fetus are unconvincing as potential causative chromosomal aberrations of the left atrial isomerism. 
  • The conclusions are inconsistent.
  • Although the manuscript describes a short case report, authors report only one reference and this is not acceptable, especially for the complexity of the issue.
  • There are several grammatical errors.

Author Response

Dear reviewer,

Thank you for taking the time to evaluate our manuscript.

We consider your comments very valuable and we have modified our manuscript according to them, as follows:

You stated: Due to the fact that heterotaxy syndrome is a very complex condition, characterized by severe cardiovascular malformations, the authors describe only the morphology of the atrial appendages and the presence of a common atrium. One of the landmarks of the left isomerism is the interruption of hepatic portion of the inferior vena cava with azygos continuation: the authors do not describe the inferior vena cava at all. Moreover, the authors do not describe the ventricular loop (L- o D-loop?), the atrioventricular connection, the ventriculoarterial connection or the presence of other structural malformations of the heart.

Reply: Your statement is correct and accurate. We have added specifications and descriptions of all the aspects of the cardiovascular findings in our affected fetus: the ventricular loop, the atrioventricular connection and we have specified that because the enlarged atrium, the drainage of the pulmonary veins were detected to be positioned lower in the left atrium, as well as the superior vena cava that drains medianly.

You stated: Line 74. The sentences: “Superior vena cava draining medianly, the pulmonary veins that drain in the lower segment of the atrium” are confusing. The anatomic description is very imprecise.

Reply: We agree and have expanded the descriptions as follows: Line 138. Superior vena cava draining medianly, the pulmonary veins that drain in the lower segment of the atrium due to the enlarged one common atrium. The extracardiac findings support and includes the left atrial isomerism in a more complex antenatally diagnosed syndrome, revealing  two bilobar lungs ( fig 5-a), dysmorphic facies, low inserted ears, nuchal fold edema and small kidneys placed in the iliac fossa and a present left sided normal spleen.   .  Genetic tests of the parents did not reveal any translocations of the chromosomes 18 and 20 when FISH analysis has been assessed. The diagnostic of the affected chromosomes using the microarray test analysis, corroborated with the antenatally detected structural abnormalities using serial ultrasound examinations, improves the genetic counseling, leading to a better assessment of the prognostic of the fetus and the recurrence rate for siblings.

You stated: Line 109. The link between mutations in the primary ciliary dyskinesia genes and heterotaxy syndrome should be expanded.

Reply: We have expanded the link between the ciliary dyskinesia syndrome and fetal heterotaxy: line 209-214

You stated: Lines 112-118. The sentences reported in these lines are very confusing and it is very difficult to understand the meaning.

Reply: We agree and have modified in the text.

You stated: The 18q deletion and 20p duplication of the fetus are unconvincing as potential causative chromosomal aberrations of the left atrial isomerism. 

Reply: Your statement is consistent, but given the low number of 20p duplication reported, it is difficult to evaluate the possible link between left atrial isomerism and the 18q deletion and 20p duplication. Our case presented these modifications and could be the starting point for further studies in this direction.

You stated: The conclusions are inconsistent.

Reply: We agree and have modified in the text.

You stated:  Although the manuscript describes a short case report, authors report only one reference and this is not acceptable, especially for the complexity of the issue.

Reply: We agree and we have added more references.

You stated: There are several grammatical errors.

Reply: We agree and have revised the text of the manuscript accordingly.

We would like to thank you very much for your recommendations. We have found them to be extremely accurate and valuable, and therefore we consider that modifying the manuscript according to your specifications has increased its value.

Best regards,

Ligia Balulescu

Reviewer 3 Report

The case report by Ilian et al describes a 26 week fetus with left atrial isomerism, the genetic work-up and the post mortem examination. The clinical descriptions and the pictures are of good quality and given the rarity of this phenotype adds to the literature. I only have some minor comments.

Minor

The English can be improved to make it easier to read, at times its “stiff”

What is the official annotation of the genetic defect according to the microarray? This is important as it helps understand exactly where the deletion and duplication took place. Preferably using hg19 or hg38 location information

Are there any other reports of cases with this exact or partly overlapping genotype? What was their phenotype? This should be discussed as it allows to put this case in context of the literature. Even if none are reported this is important information to state.

It’s unclear what type of test was initially performed that gave the normal karyotype in the fetus. It just says pcr test; please expand the method or use a reference here

Spelling apsrenia line 38

Author Response

Dear reviewer,

Thank you for taking the time to evaluate our manuscript.

We consider your comments very valuable and we have modified our manuscript according to them, as follows:

You stated: The case report by Ilian et al describes a 26 week fetus with left atrial isomerism, the genetic work-up and the post mortem examination. The clinical descriptions and the pictures are of good quality and given the rarity of this phenotype adds to the literature. I only have some minor comments.

 Reply: We very much appreciate your king remarks and the interest in our case report.

You stated: The English can be improved to make it easier to read, at times its “stiff”

Reply: We agree and have had the text revised by a native English speaker.

You asked: What is the official annotation of the genetic defect according to the microarray? This is important as it helps understand exactly where the deletion and duplication took place. Preferably using hg19 or hg38 location information

 Reply: We have added the requested information in the text – lines 123-128.

You asked:  Are there any other reports of cases with this exact or partly overlapping genotype? What was their phenotype? This should be discussed as it allows to put this case in context of the literature. Even if none are reported this is important information to state.

Reply:  We agree and have added the requested information in the text: lines 244-256

You stated: It’s unclear what type of test was initially performed that gave the normal karyotype in the fetus. It just says pcr test; please expand the method or use a reference here

Reply: We agree and have added the requested information in the text: lines 123-128.

You stated: Spelling apsrenia line 38

Reply: We appreciate your vigilant remark and have corrected the misspelled word in the text.

We would like to thank you very much for your recommendations. We have found them to be extremely accurate and valuable, and therefore we consider that modifying the manuscript according to your specifications has increased its value.

Best regards,

Ligia Balulescu

Round 2

Reviewer 2 Report

In the revised version of the manuscript the authors have addressed most of the remarks that I raised. However, the authors stated that they added all the cardiovascular findings of the fetus but I did not recognize any specifications about ventricular loop, the atrioventricular and ventriculoarterial connections or the relationship of the great arteries one another. Due to the well-known cardiovascular complexity of left isomerism, and of heterotaxy in general, it is very important to provide readers a complete description of the heart and great arteries. If they complete the anatomical description of the cardiovascular findings of the fetus, in my opinion the paper can be considered for publication.

Author Response

Dear reviewer,

Thank you for taking the time to reevaluate our manuscript.

We very much appreciate the interest in our case report.

You stated: In the revised version of the manuscript the authors have addressed most of the remarks that I raised. However, the authors stated that they added all the cardiovascular findings of the fetus but I did not recognize any specifications about ventricular loop, the atrioventricular and ventriculoarterial connections or the relationship of the great arteries one another. Due to the well-known cardiovascular complexity of left isomerism, and of heterotaxy in general, it is very important to provide readers a complete description of the heart and great arteries. If they complete the anatomical description of the cardiovascular findings of the fetus, in my opinion the paper can be considered for publication.

Response: We kindly appreciate your suggestions and have carried out the following modifications regarding the anatomical description of the cardiovascular findings of the fetus: lines 128-147:

The ultrasound cardiac findings were confirmed by a pediatric cardiologist and a diagnostic amniocentesis was performed.

Regarding the cardiac relations and the segmental approach we noted an “A” situs ambiguous of a left atrial isomerism with left atrial appendage, a rightward rotation with a D-loop of the ventricular mass (right hand pattern ventricular topology “D”), together with “S” normally related arteries (“ADS”) [6, 7].  Mixed biventricular atrioventricular connections guarded by two normal AV valves, the mitral valve at the level of the left sided heart and tricuspid valve placed at the level of the right side of the heart. Concordant ventriculoarterial connections (in biventricular heart) and normal position of the heart within the chest, with left sided ventricular mass and a left sided cardiac apex were detected. A normal relationship of great vessels-descending aorta, with a left sided descending aorta and a right sided inferior vena cava in relation with the spine, as well as a normal relation regarding the aortic orifice, situated right and posterior with respect to the pulmonary valve. A particularity of the  superior vena cava that was placed above the plane of the 3 vessel view is noted (Figure 2 A), a deviation determined by the enlarged right sided atrium. All four systemic veins drain in the left atrium with the anatomically normal systemic veins draining into the right sided atrial chamber that possesses a morphologically left appendage. Within the heart,  no atrial septum was noted, nor the presence of the foramen ovale. We noted the presence of the gallbladder, one anatomically normal spleen, normal splenic artery and a normally structured and positioned fetal liver in the right side of the abdomen.

We would like to thank you very much for your recommendations. We have found them to be extremely accurate and valuable, and therefore we consider that modifying the manuscript according to your specifications has increased its value.

Best regards,

Ligia Balulescu